# Ovariopexy—Before and after Endometriosis Surgery

**DOI:** 10.3390/biomedicines8120533

**Published:** 2020-11-25

**Authors:** Juhi Dhanawat, Julian Pape, Damaris Freytag, Nicolai Maass, Ibrahim Alkatout

**Affiliations:** Department of Gynecology and Obstetrics, University Hospitals Schleswig-Holstein, Campus Kiel, Arnold-Heller Str. 3, Building 24, 24105 Kiel, Germany; JulianMaria.Pape@uksh.de (J.P.); Damaris.Freytag@uksh.de (D.F.); nicolai.maass@uksh.de (N.M.)

**Keywords:** ovariopexy, endometriosis, surgical access, technical feasibility, retro-peritoneum, postoperative adhesion, laparoscopy

## Abstract

Endometriosis surgery is often very challenging. Key to complete resection of endometriosis is access to the retroperitoneum. Endometriosis can involve the ureter and uterine vessels, and ovary on the lateral pelvic wall makes retroperitoneal access difficult. Primary and post-surgical adhesions prevalence in endometriosis is very high. Ovariopexy, transposition of ovaries temporarily, is done for better surgical access and to reduce postoperative adhesions. We concluded that although limited evidence, ovariopexy is an excellent tool to aid endometriosis surgery and prevent postoperative adhesions. It is cost effective, simple and complication rate almost nil. More robust trials are required to substantiate evidence for its impact on preventing postoperative adhesions and its effect on fertility. In this review, we describe our technique of ovariopexy supplemented with a video, with the aim to put light on this useful and important technique, which is beneficial both for surgeons and patients.

## 1. Introduction

The pelvis is a closed space. Surgery deep in the pelvis and accessing retroperitoneal pelvic structures requires free space intra-peritoneally during laparoscopic surgery. The minimally invasive technique, with limited flexibility, often faces the problem of a freely manageable surgical situs with the use of limited assistance. The lateral pelvic wall and pouch of Douglas, usually accessed by gynecologists, urologists, and bowel surgeons, require this free space.

Endometriosis, which affects these areas, largely has a worldwide prevalence from 0.8% to 28.6% [1]. It is the second most common benign gynecological disorder [2]. A recent review on the prevalence of endometriosis found it to be 64% in adolescents who have undergone laparoscopy [3]. It has a prevalence of 40–60% in women with dysmenorrhoea, 21–47% in women with sub-fertility, and 71–87% with pelvic pain [4]. With such a high prevalence, endometriosis has been described as nothing short of a public healthcare emergency [5].

Many classifications have been proposed to classify endometriosis, but none are perfect. The most commonly used classification worldwide is the revised American Society of Reproductive Medicine classification (rASRM).

According to rASRM classification, endometriosis is graded into four stages depending on the size, depth, presence of flimsy or dense adhesions, obliteration of the pouch of Douglas or not, and site of the endometriotic lesion in the peritoneum or ovary [6]. This is the only classification thatdocuments the complete adhesion status of the disease [7]. Adhesions can be associated with superficial as well as deep infiltrating endometriosis. Stages III and IV may have deep infiltrating endometriotic nodules with endometriotic ovarian cyst with dense adhesions, which usually involves ovaries, the pouch of Douglas, lateral pelvic wall, and bowel [8]. Deep infiltrating endometriosis constitutes about a total of 48% of all endometriosis cases, hence, the prevalence of adhesions in endometriosis is also very high [6]. According to the world society endometriosis consensus 2017, the Enzian classification system along with rASRM classification should be used to give complete description of the disease extent, as this system describes the deep infiltrating endometriosis better, but adhesions are described to the full extent in rASRM classification [9].

Diagnosis of endometriosis is primarily made by history, physical examination, ultrasound, and magnetic resonance imagining (MRI). However, none of the imaging techniques have a 100% sensitivity or specificity [5]. The gold standard for diagnosis and treatment of endometriosis is essentially surgery, more often treated by laparoscopy than open [10,11]. One major problem encountered during laparoscopic surgery of endometriosis is access to the endometriotic lesions or nodules in the ovarian fossa, a deep infiltrating endometriotic nodule in the uterosacral ligaments and pouch of Douglas. The peritoneal cavity present behind the uterus should be free to have a clear vision of such sites. To have a clear view of such sites, the ovaries and tube should be temporarily put away. Repeated falling of tubes and ovaries during surgery hampers the vision of the surgical site, decreases access to such sites, increases chances of error, where very often these nodules involve deeper important structures like ureter, uterine artery, and hypogastric nerves. Likewise, urologist need the adnexa away, to access the pelvic ureter and pelvic vessels, and bowel surgeons when working on the sigmoid colon and rectum.

Adhesions in endometriosis are acquired adhesions, which can be either postoperative and inflammatoryadhesions. Common to both is that they are caused by damage to the peritonealmesothelium influenced by ischemia, inflammatory processes, surgical trauma, or radiation [12]. During the subsequent wound healing and activation of the coagulation cascade, there is an imbalance between fibrin production and breakdown, which causes fibroblasts to secrete more cytokines that promote collagen synthesis. This leads to excessive formation of scar tissue, which allows two tissues to adhere to one another [13]. Bacterial infections can reduce fibrinolysis in the peritoneum, promoting adhesions. The genetic-epigenetic pathophysiology of endometriosis is considered to be influenced by infections, with women with endometriosis having more risk of genital tract infections. Endometriosis is known to be associated with upper genital tract infections and peritoneal infections, which could be a cause of increased adhesions associated with this disease [14]. Endometriosis is primarily associated with inflammatoryadhesions. If endometriosis cannot be removed from the peritoneum in good time, this leads to an inflammatory reaction, which leadstoadhesions between two tissues [15]. Considering that endometriosis causes adhesions, postoperatively, it is the emergence of new adhesions thatis a dilemma and needs to be reduced. Menzies et al. found abdominal adhesions in 93% of the previously operated patients [16]. Second look laparoscopy revealed that 75% of patients had adhesions after laparotomy and 15% after laparoscopy [13].

The solution to the above problems is ovariopexy, repositioning of the ovary temporarily by fixing it to another structure in its vicinity, maintaining the ovarian supply from both the ovarian ligament and infundibulopelvic ligament. Ovarian transposition is a similar technique, but the ovary is fixed to some other structure, usually the upper anterior abdominal wall, while separated from the uterus. In this case, the blood supply to the ovary is only maintained by the infundibulopelvic ligament [17]. Ovariopexy and ovarian transposition havebeen used for a long time in young women undergoing radiation to remove the ovary from the radiation site for fertility preservation [17,18]. It was done by McCall et al.for cervical cancer patients by laparotomy after radical hysterectomy for fertility preservation [19]. Now it is being done laparoscopically as well as in robotic surgeries [17]. Ovariopexy has also been used for adnexal torsion, where the ovary is fixed to the round ligament, or the ovarian ligament is shortened, and the ovary is fixed to it to prevent further torsions [20]. Therefore, if a little change is done in the position of the ovary surgically, it can also be beneficial in other cases like endometriosis, large ovarian cystectomy to reduce postoperative adhesions and gain access to deeper pelvic structures. This technique can also benefit urologist, uro-gynecologist, and bowel surgeons who need access to the pelvis. There are only a few studies that mention ovariopexy as a benefit for access to the lateral pelvic wall in endometriosis surgery [21,22,23,24] and, there are some studies thathave shown a benefit in preventing postoperative adhesions [25,26,27,28].

By this article, we aim to increase awareness of an already established simple technique of ovariopexy, as a useful adjunct to endometriosis surgery, before and after the procedure.

## 2. Technique of Ovariopexy

After gaining access to the peritoneal cavity laparoscopically (Figure 1), the cavity is thoroughly investigated for endometriotic cysts, spots, and adhesions. If an endometriotic ovarian cyst is found, it is excised. If endometriotic spots or nodules are found underneath the ovary on the ovarian fossa, or uterosacral ligaments, the ipsilateral ovary is suspended to the anterior abdominal wall (Figure 2). A straight needle with a single monofilament non-absorbable thread is passed through the lower anterior abdominal wall. Prior to insertion of the needle, the inferior epigastric vessels should be located to avoid injury and hematoma formation. The needle is retrieved intra-corporeally by grasping it using a laparoscopic needle holder. It is then passed from the medial side of the ovary through the ovary into the abdominal wall close to the point where it was introduced. At this step also, it is essential to pay attention to the inferior epigastric vessels once more. The needle is then pulled out from the abdominal wall using a hemostat forceps. The two ends of the thread are tied extra-corporeally over a compressor gauze on the abdominal wall, with some tension, thus suspending the ovary to the abdominal wall with the medial side of the ovary apposed to the peritoneum of the anterior abdominal wall (Appendix A).

Depending on the in-situ findings, unilateral or bilateral ovaries can be suspended.

After the procedure is complete, the ovarian suspension is loosened to maintain a gap of 1–2 cm between the ovary and the abdominal wall to prevent adhesions. This suspension is maintained for 5 days, and sutures are cut on the day 5th-day post-surgery (Figure 3).

Another illustration after endometriosis resection with the important anatomical structures shown (Figure 4). Vessels in the abdominal wall have to be kept in mind before doing an ovariopexy to prevent puncture of it [29]. Comfortable access to the lateral pelvic wall is really important to prevent damage to major structures like the ureter and uterine artery, a very common site for superficial and deep endometriosis.

Most authors perform ovariopexy by the above technique as we do [21,22,25,26,27,28,30]. Two authors suspended the ovary to the ipsilateral round ligament using absorbable polyglactin suture, vicryl [31,32]. There are theoretical chances of the ovary getting adhered to the round ligament, which may affect fertility potentially naturally by altering the normal anatomy, as well as it could make oocyte retrieval difficult. Nevertheless, both authors have shown reduced postoperative adhesions with the above technique.Chapman et al havedone ovariopexy in another way wherein both the ovaries are suspended by one suture anterior to the uterus on the abdominal wall [23]. This technique changes the course of the ureter and cannot be used to prevent postoperative adhesions.

Most authors used non-absorbable polypropylene monofilament suture (Prolene 0) or non-absorbable braided polyester suture (Mersuture 0) [25,26,27,28,30,33] for ovarian suspension on the abdominal wall. Recently, Abuzeid et al. performed the same technique, suspended the ovary to the fascia of the abdominal wall using absorbable 3-0 plain catgut suture [34]. This technique would have to be done solely for adhesion prevention and will not aid in access to the ovarian fossa for endometriosis surgery. Since an absorbable suture was used, removal of the suture was not required and theoretical risk of infections is absent. They subsequently published a study thatshowed similar efficacy when absorbable or non-absorbable sutures are used with no evidence of decreased ovarian reserve in both. He also concluded, absorbable and non-absorbable sutures can reduce postoperative adhesions between the ovary and ovarian fossa [22].

The suspension time of the ovaries is debatable. Hoo et al. suspended the ovary for 36 to 48 h and did not find this technique useful to reduce postoperative adhesions [30]. Trehan et al. concluded that ovary suspension time being 36–48h in Hoo et al.’s study was inadequate. According to him, the ovary should be suspended for 7–9 days because peritoneal healing takes 5–8 days time, and blood in the pelvic cavity takes 8 days for absorption, which is one risk factor for adhesion formation [35]. All other authors suspend ovaries for 5–7 days in accordance with peritoneal healing [22,25,26,27]. Technique of ovariopexy done by many authors over the years is summarized in Table 1.

## 3. Counseling

As much as the surgery for endometriosis is challenging, counseling of patients regarding endometriosis and its treatment and implications is also very difficult. Prior to endometriosis surgery, every patient should be explained about the extent of her disease as estimated by her symptoms and pre-operative imaging techniques. If lesions of the ovary, ovarian fossa, uterosacral ligaments, pouch of Douglas, or involvement of the bowel is suspected, thorough discussion of its implications should be explained, and that adhesions in such cases are expected, and that surgery itself can also lead to adhesions. The technique of ovariopexy should be discussed, that it would be done before the actual resection of endometriosis is done to aid in better surgery, and if required, single or bilateral ovariopexy would be left behind to prevent postoperative adhesions [36].

Postoperative adhesions occur in 50%–100% of patients [22] and can lead to chronic pelvic pain, infertility issues, dyspareunia, and in extreme cases, bowel obstructions [37]. According to a panel of European experts (anti-adhesion in Gynaecology expert group [ANGEL] and the European Society of Gynaecological Endoscopy (ESGE), all patients undergoing abdominal surgery should be counseled about postoperative adhesions and its risks, multiple ways to reduce the adhesions. Surgeons should have a practice guideline which they follow to prevent adhesions, and the same should be discussed with the patient [36].

Usually, none or only mild discomfort is felt after ovariopexy. The patient will be discharged to go home as per hospital protocol and according to her postoperative recovery. The sutures can be cut by the women’s local gynaecologist without leaving any marks on the skin if discharged earlier than 5 days. This procedure carries a very low risk of complications, almost none [4].

## 4. Advantages of Ovariopexy: Surgeon’s Way to Maximize Surgical Outcome

### 4.1. Access to Surgical Field

If a good surgical technique is followed, intra-operative and postoperative complications can be minimized significantly. Endometriosis surgery is challenging, even in the hands of an expert. Distortion of anatomy, dense adhesions, the involvement of important structures like the ureter, bowel, and site of endometriotic nodules with deep infiltration into the pelvic structures makes this surgery difficult, time consuming, bloody operative field, thereby frustrating the surgeon, which could lead to an incomplete surgery or more chances of injury and complications. Adnexa hinders access to the lateral pelvic wall [23].

Hence, by doing ovariopexy, either unilateral or bilateral, depending on in-situ findings, will aid the surgeon in multiple ways.

Better visualization and access to the ovarian fossa, uterosacral nodules, and pouch of Douglas decreases the risk of injury, and more radical surgery can be done with ease. Less number of are trocars used, relieves the burden of the assistant surgeon, and additionally optimizes the use of instruments and assistant.Potential damage of mesovarium and ovary by a grasper is eliminated, and vision hindrance by the grasper itself is avoided by doing an ovariopexy [23].

Ovariopexy for better surgical access has been described earlier, similar to ours by Cutner et al. [24] and a slightly different technique by Chapman et al. [23], wherein both advocate ovariopexy for technical feasibility in terms of surgery.

Some other surgical techniques like uteropexy and bowelpexy can be done along with ovariopexy to improve surgical access of the pelvis. Temporary displacement of the uterus to the anterior abdominal wall will further ease the availability of the pouch of Douglas and uterosacrals [38]. Fixing the epiplocae of the bowel to the lateral abdominal wall will remove the bowel from the pelvis, which is especially very beneficial in obese women and women in whom the trendelenberg position or increased intrabdominal pressure is contraindicated. The fixations should be removed immediately after the completion of surgery. This, in our experience, improves surgical access to the pelvis and ease with which the surgery is done.

### 4.2. Reduction of Postoperative Adhesion

Various studies have looked into the mechanism of adhesion formation. Surgical trauma to the peritoneum disrupts the normal healing process. Fibrin deposition occurs within the first 3h of injury, and fibrinolysis occurs after 72 h. This fibrinolysis is important for the normal healing process. If fibrinolysis does not occur, it leads to adhesions [39]. Awareness of postoperative adhesions is lacking amongst surgeons, and only a few used some method to prevent adhesions [40,41,42].

Postoperative adhesions are detrimental for surgeons as well as patients. It increases surgery time, risk of injury to bowel and bladder, and increases financial burden also with an estimated cost of adhesion-related admission to be 24.2 million Euros after 2 years and 95.2 million Euros after 5 years of surgery [37]. Chronic pain and small bowel obstruction due to adhesions are major causes of readmissions [39]. Post operative adhesions are inevitable after any abdominal surgery but various recommendations have been made to prevent or reduce it. One needs to follow good surgical techniques like gentle handling of tissues, thorough hemostasis, minimize ischemia, reduce cautery time, and aspirate aerosolized tissue after cautery, excise endometriotic tissue rather than coagulation, reduce surgery time, and reduce pressure and duration of pneumoperitoneum [36].

Postoperative adhesions can be reduced using various barrier agents approved by FDA like seprafilm, interceede, Gore-Tex surgical membrane, but all have limited evidence of their efficacy [36,39]. A recent study shows the benefit of using a product thatis plant-based (modified polysaccharide dry 4 field^®^PH-PlanTec Medical GmBH, Bad Bevensen, Germany). This powdered agent, when sprinkled on tissue, is a hemostatic agent, and when mixed with a little saline, becomes an anti-adhesive agent. The authors have shown an 85% reduction of postoperative adhesions after use of this agent [43]. The cost of these anti-adhesion agents hasto be taken into account, which becomes all the more important in a low resource country. The use of these substances in clinical practice is mostly unusual, and the health care system usually does not adequately reimburse the costs for anti-adhesion preparations.

Ovaries are the commonest site for adhesion development due to their approximation with the lateral pelvic wall, fallopian tube bowel, omentum, and the pouch of Douglas. The coelomic epithelium covering the ovary is fragile, leading to more adhesion formation [44]. If, after endometriosis surgery, the ovaries are kept away from the raw peritoneal surfaces for a minimum time period of 5–7 days, the formation of adhesions can be prevented. By doing an ovariopexy, we achieve the above. This process does not increase the cost or time duration of surgery, it is simple and very easy to learn. Reduction of postoperative adhesions after ovariopexy is equivocal, with reductions of adhesions seen in 40–80%. Oubha et al. had significant postoperative dense adhesions between the ovary and lateral pelvic wall only in 33.3% patients (4/12), with 41.6% patients (5/12) adhesion free, and 25% patients (3/12) having flimsy adhesions to the ipsilateral tube [28]. Carbonnelet al. had a reduction in 50% of patients (19/38) [27], while Abuzied et al. had an absence of adhesions in 80% of patients [22]. Pellicano et al. reported a postoperative adhesion formation of 40.7% [31]. Hoo et al. showed less than 50% postoperative reduction in adhesions, but he suspended the ovaries for 48 h only, which could be the reason for inadequacy [30]. Serrachioliet al. reported significantly low amountsof adhesions, with the uterus and bowel in the group whose ovaries were suspended compared tothe control (46.7% vs. 77.3% for adhesions with the uterus and 26.7% vs. 68.2% for bowel adhesions, respectively) [21]. Although results are conflicting, this technique looks promising. Two systematic reviews done also conclude that ovarian suspension is an effective surgical procedure thatwill reduce postoperative adhesions [4,45]. Although second look laparoscopy confirms postoperative adhesions, ultrasound has been shown to predict adhesions with 80% sensitivity and specificity [7]. Okaro et al. had a good correlation between ovarian mobility by laparoscopy and transvaginal ultrasound [46]. Therefore, robust trials can be done non-invasively to predict more convincing results of ovariopexy in reducing postoperative adhesions non-invasively.

## 5. Effect of Ovariopexyon Fertility

There are a few authors who have studied the effect of ovariopexy on fertility. Dehbashiet al. followed up his patients (50) 2–4 years after surgery. Infertility was reported in 38 patients (17 in the suspended group and 21 in the control). All infertile patients received assisted reproductive technique (ART). Chemical pregnancy rate and clinical pregnancy rate did not differ in both groups [25]. Carbonnel et al. in theirretrospective study, included 218 patients with 336 ovaries suspended. 132 infertile women tried to conceive, 105(79.5%) answered the questionnaire. 21(36%) patients conceived spontaneously, 22(38%) after the first attempt of ART, 15(26%) after several attempts. The cumulative pregnancy rate after more than 24 months post-surgery was 76% [27]. Ouabha et al. had 53.3% pregnancy rate amongst patients who underwent ovarian suspension onaverage within 11.5 months [28]. Poncelet et al. evaluated oocyte retrieval before and after ovariopexy, and concluded there was no difference in ovarian accessibility before (96.7%) and after (96.1%) surgery. Antral follicle count and pulsatility index of suspended ovaries werenot different from contralateral unsuspended ovaries [26]. Abuzied et al. in theirstudy compared non-absorbable suture and absorbable suture in ovariopexy, concluded that both groups had similar clinical pregnancy rates (44.2% vs. 34.3%) and delivery rate (36.5% vs. 31.3%) who conceived spontaneously [22]. Interpretation of these results with respect to the ovariopexy is difficult due to various confounding factors, like reproductive age of women, male factor infertility, and women with endometriosis have low fertility potential. Ovarian suspension may have contributed to successful pregnancies by reducing adhesions and maintaining normal anatomy.

## 6. Effect of Ovariopexyon Pain

In our unpublished data, immediate postoperative pain at the ovariopexy site is nil. Patients do not require additional analgesics for the same. Abuzeid et al. also mention that none of his patients while in the hospital or after discharge, had pain different from patients undergoing operative laparoscopy for other reasons [22]. Pellicano et al. evaluated postoperative pain using a pain visual analog scale(VAS) 24 h after surgery and found no difference between the ovarian suspension and non ovarian suspension groups [31]. Seracchioli et al. evaluated pain before and 6 months after surgery using VAS, and found no difference between suspended and non-suspended groups. Pain scores improved after surgery, regardless of ovarian suspension [21]. Hoo et al. showed improved postoperative pain despite finding no benefit of ovariopexy in the reduction of adhesions [47].

## 7. Complications

Ovariopexy is a very safe procedure with very few complications cited in the literature, which could be directly caused by ovariopexy.Carbonnelet al. had two complications amongst 336 suspended ovaries (0.6%), one ovarian abscess, which was drained by posterior colpotomy, and one hemoperitoneum from multiple bleeding sites, including the suspension site, but the ovary was preserved [27]. Poncelet et al. had a 0.7% complication rate (2/279), one ovarian abscess due to Klebsiellapneumoniae, which was drained by a posterior colpotomy on day 8, and one hemoperitomeum from several bleeding sites, the ovary was preserved [26]. Hoo et al. mentionedthe potential risk of bowel strangulation in the ovariopexy suture, but it is a theoretical risk [30]. None of the authors have had this complication. Abuzied et al. mentioned suture breakage in 3 patients wherein catgut was used and ovary suspended to the fascia. The suture was replaced in all, and minor bleeding of the ovarian ligament was cauterized by bipolar energy [22]. In our experience, we had only one complication with ovariopexythus far, hematoma in the subcutaneous tissue. While retrieving the needle from the abdomen into the abdominal wall, the inferior epigastric vein was punctured, leading to hematoma, which was controlled immediately by coagulation. Careful visualization of inferior epigastric vessels while inserting as well as retrieving the needle from the abdominal wall is essential.

The above factors in relation to ovariopexyhave been summarized in Table 2.

## 8. Conclusions

Ovariopexy is a simple surgical tool that can be utilized in complicated and challenging surgeries like endometriosis for better surgical access and radicalization of surgery. The time for suspension of the ovary is around 5–7 days post-surgery. Absorbable or non-absorbable sutures can be used. It has shown promising results in preventing postoperative adhesions, but more randomized clinical trials are required. It has a negligible complication rate, does not increase pain, and is cost-effective. It may or may not directly affect fertility, but it does not have a negative effect on fertility or oocyte retrieval. Lack of awareness of this technique is probably why it is underutilized. Ovariopexy is a great technique, which compliments a surgery outcome.

## 9. Future Perspective

Clinical trials regarding surgical ease after ovariopexy can be done by asking beginners and advanced surgeons to rate the technique in the form of a questionnaire. This will validate and give estimates ofthe advantages of this technique.

More robust randomized control trials are needed to evaluate the effect of ovariopexy on postoperative adhesions. It has been well documented that ultrasound can detect adhesions in the hands of an experienced sonologist. Now, large trials can be conducted non-invasively to detect postoperative adhesions post ovariopexy in endometriosis surgery. In our center, we will be conducting a randomized control trial to detect postoperative adhesions in all patients undergoing an ovariopexy.

Clinical trials should be done with regards to fertility and clinical pregnancy rate after ovariopexy. Multiple factors like male infertility, tubal and uterine factors, age, previous medical and surgical history impacting fertility should be taken into account while conducting these trials to give unbiased evidence regarding the effect of ovariopexy on fertility.

## Figures and Tables

**Figure 1 biomedicines-08-00533-f001:**
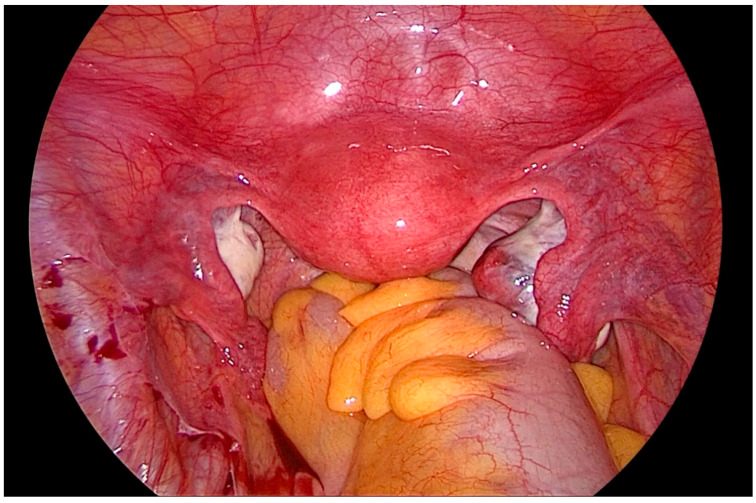
Laparoscopic view of the pelvis, depicting the ovaries hampering vision and access to the lateral pelvic wall.

**Figure 2 biomedicines-08-00533-f002:**
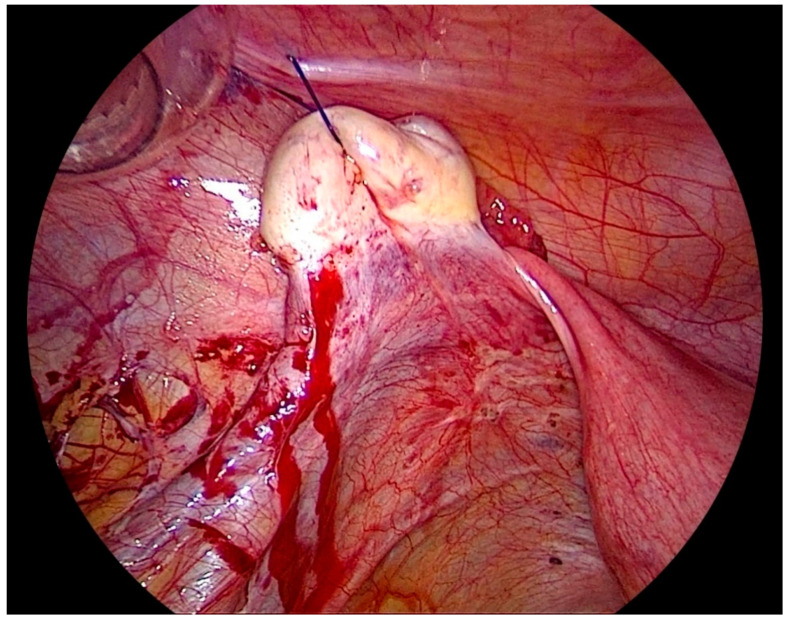
Laparoscopic view of the lateral pelvic wall on the left side after ovariopexy with endometriosis easily visualized and accessible.

**Figure 3 biomedicines-08-00533-f003:**
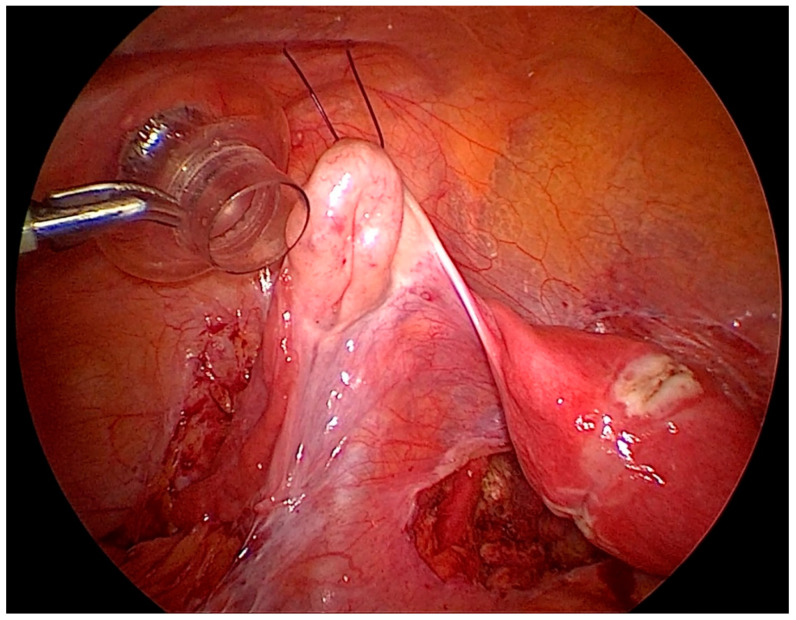
Left side ovary suspended to the anterior abdominal wall, with dissection of the left lateral pelvic wall.

**Figure 4 biomedicines-08-00533-f004:**
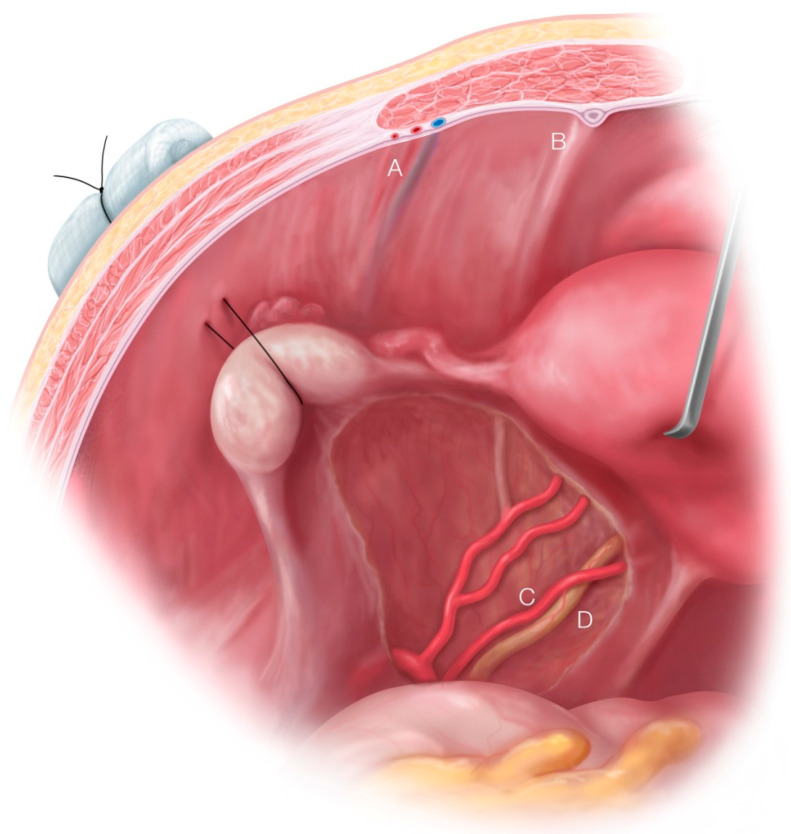
Illustration showing left suspended ovary to the anterior abdominal wall after endometriosis surgery with important anatomical structures. (**A**) Inferior epigastric vessels in the lateral umbilical fold. (**B**) Medial umbilical ligament. (**C**) Uterine Artery. (**D**) Ureter.

**Table 1 biomedicines-08-00533-t001:** Summary of Ovariopexy technique.

Author	Year	Ovariopexy Site	Suture	Day of Suspension Removal
Redwine	2001	ipsilateral round ligament	vicryl 3-0	N/A
Abuzeid	2002	anterior abdominal wall *	polypropylene 3-0	5th–7th
Oubha	2004	anterior abdominal wall *	prolene 3-0	4th
Chapman	2007	Both ovaries together, in front of the uterus, on the anterior abdominal wall	Vicryl 2-0	Immediately ater surgery
Carbonnel	2011	anterior abdominal wall *	prolene 0,mersuture	5th
Hoo	2011	anterior abdominal wall *	prolene	36–48 h
Poncelet	2012	anterior abdominal wall *	prolene-0	5th
Seracchioli	2014	anterolateral abdominal wall *	vicryl 2-0	no
Pellicano	2014	ipsilateral round ligament	vicryl rapid 2-0	N/A
Abuzeid	2018	group1- fascia of anterior abdominal wall *; group 2-anterior abdominal wall *	group 1-plain catgut 3-0; group 2-nylon 3-0	group 2–5th–7th
Dehbashi	2019	anterior abdominal wall *	prolene	7th

* Each ovary suspended separately.

**Table 2 biomedicines-08-00533-t002:** Summary of various factors in relation to ovariopexy.

Author	Year	Postoperative Adhesions Formation	Pain at Surgical Site	Fertility (Clinical Pregnancy Rate)	Complications
Redwine	2001	0%(3 patients)			None
Abuzeid	2002	20% (1/5)(minimal adhesions)		45%(9/20)	None
Oubha	2004	58.3%(7/12)(flimsy+mild adhseions)		53.3%(8/15)	None
Carbonnel	2011	50%		55%(58/105)	0.60%
Hoo	2011	68.8%(11/16)			None
Poncelet	2012			No differencein oocyte retrieval rate, pulsatility index, and antral follicle count between suspended and non-suspended ovaries.	0.70%
Seracchioli	2014	37.7% vs. 77.2%(17/45 vs. 34/44)(moderate and severe adhesion)(suspended vs. non-suspended)	No difference between the suspended and non suspended site		None
Pellicano	2014	33.3%(8/24)vs. 80.8%(21/26) (suspended vs. non-suspended)	No difference between the suspended and non suspended site		None
Abuzeid	2018	0%(both groups)		41.00%	None
Dehbashi	2019	18%vs. 76%(9/50 vs. 38/50)(moderate adhesions; suspended vs. non-suspended)		19% vs. 22%(suspended vs. non-suspended)	None

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
