# Peer review of "Ovariopexy—Before and after Endometriosis Surgery"

_biomedicines, 2020, doi:10.3390/biomedicines8120533_

Round 1

Reviewer 1 Report

Paper is well structured and well-written. Wide spectrum of literature has been reviewed. 

The work for a review is excellent, but the method described by the authors does not fit to the criteria of review. It should not be involved, or should be added the exact results of their study and change the form to an original article.

Reviewer 2 Report

The authors have addressed a known and rather innovative surgical technique, which has been less focused on, in current times. Ovariopexy is a very important and useful tool in laparoscopic surgeries, and more so in endometriosis surgery.

The review article, is written well with a good focus on important factors regarding ovariopexy especially in relation to endometriosis. A thorough research on medical databases such as pubmed/google scholar has been made. All relevant articles up till August 2020 have been considered.

The results of the research is summarised adequately under the following:

Technique of ovariopexy

Advantages- good surgical access, reduction of postoperative adhesions

Effect on fertility

Effect on pain

Complications

Furthermore, the systematic technique of ovariopexy is well explained, which can be easily understood and adopted by surgeons worldwide and of all surgical fields. It is supplemented with high quality images and specially crafted illustrations, along with anatomical descriptions relevant to it. Comparison with other techniques described in literature has been done and tabulated with respect to site of ovariopexy, suture material used and number of days of suspension. All other results have also been tabulated, painting a clear picture in the readers mind about ovariopexy.

Based on the results of the research, I have some minor suggestions:

  1. The authors should highlight, what are the possible research areas needed further to have a more conclusive evidence of effect of ovariopexy in relation to endometriosis.
  2. If any other surgical technique is available to supplement ovariopexy, for good surgical access of endometriosis.

Overall I think, it is a very well written review on ovariopexy, highlighting not only reduction of post operative adhesions, but also good surgical access. Although it is a very important and useful technique, this technique has not gained enough importance and this article beautifully focuses on its different aspects.

Therefore, in this special issue on endometriosis, I would strongly recommend to accept this article with high priority. Furthermore, this article will usefully compliment the molecular articles in the journal issue and help to complete the overall picture of endometriosis.

I therefore recommend to accept this article, after the minor revisions have been made respectively.

Round 2

Reviewer 1 Report

Acceptable in this form. In an original article own data of the technique and authors should be published